# *LazyLLM*: Dynamic Token Pruning for Efficient Long Context LLM Inference

## Abstract

The inference of transformer-based large language models consists of two sequential stages: 1) a *prefilling* stage to compute the KV cache of prompts and generate the first token, and 2) a *decoding* stage to generate subsequent tokens. For long prompts, the KV cache must be computed for all tokens during the *prefilling* stage, which can significantly increase the time needed to generate the first token. Consequently, the *prefilling* stage may become a bottleneck in the generation process. An open question remains whether all prompt tokens are essential for generating the first token. To answer this, we introduce a novel method, *LazyLLM*, that selectively computes the KV for tokens important for the next token prediction in both the *prefilling* and *decoding* stages. Contrary to static pruning approaches that prune the prompt at once, *LazyLLM* allows language models to dynamically select different subsets of tokens from the context in different generation steps, even though they might be pruned in previous steps. Extensive experiments on standard datasets across various tasks demonstrate that *LazyLLM* is a generic method that can be seamlessly integrated with existing language models to significantly accelerate the generation **without fine-tuning**. For instance, in the multi-document question-answering task, *LazyLLM* accelerates the *prefilling* stage of the LLama 2 7B model by $2.34\times$ while maintaining accuracy.

## 1 Introduction

Standard prompt-based LLM inference has two sequential stages: *prefilling* and *decoding*, as shown in Figure 1. During the *prefilling* stage, the model computes and saves the KV cache of each token from the prompt, and predicts the first token. We refer to the time taken during *prefilling* stage as "time-to-first-token" (*TTFT*). Following the *prefilling* stage is the *decoding* stage, where the model reuses cached KVs to decode the next token iteratively until the stop criteria are met.

During the *prefilling* stage, all tokens from the prompt are used by all transformer layers. For long prompts, *TTFT* could be slow because state-of-the-art transformer-based LLMs are both deep and wide (Pope et al., 2023; Kim et al., 2023; Aminabadi et al., 2022), and the cost of computing attention increases quadratically with the number of tokens in the prompts. For instance, Llama 2 (Touvron et al., 2023), with 7 billion parameters, stacks 32 transformer layers with a model dimension of 4096. In this scenario, *TTFT* requires $21\times$ the walltime of each subsequent decoding step, and accounts for approximately 23% of the total generation time on the LongBench benchmark[1] (Bai et al., 2023). Therefore, optimizing *TTFT* is a critical path toward efficient LLM inference (NVIDIA, 2024).

While optimizing LLM inference is an active area of research, many methods (Leviathan et al., 2023; Cai et al., 2024; Zhang et al., 2024; Bhendawade et al., 2024; Li et al., 2024) have focused on improving inference speed during the *decoding* stage. Yet, there is little attention given to improving *TTFT*. We note that some compression-based works implicitly improve the *TTFT* by reducing the size of LLMs (Frantar et al., 2022; Sun et al., 2023; Ma et al., 2023). However, an orthogonal line of research(Li et al., 2023; Jiang et al., 2023; Dao et al., 2022) investigates how *TTFT* can be improved given a static transformer architecture. Within this line of research, a natural question arises: Are all prompt tokens essential for generating the first token?

---

[1]The average LongBench prompt length is 3376 tokens and the average generation length is 68 tokens.

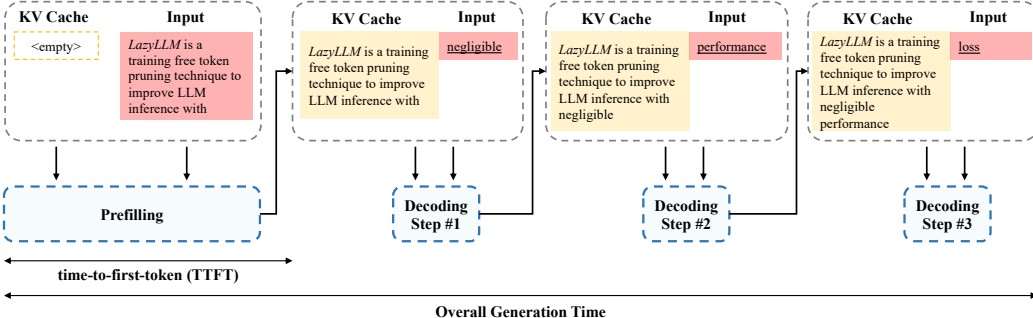

Figure 1: Prompt-based LLM inference can be divided into two sequential stages: *prefilling* and *decoding*. For long prompts, the first token generation during *prefilling* stage could be slow. As an example, for Llama 2 7B model (Touvron et al., 2023), on average, the time to generate the first token requires $21\times$ the walltime of each subsequent decoding step and accounts for $23\%$ of the total generation time in the LongBench benchmark.

LLM profiling on the LongBench benchmark (Bai et al., 2023) in Figure 2 reveals that the attention scores of input tokens w.r.t. to the first generated token are very sparse, indicating that many tokens in the input prompt are redundant and can be removed without affecting the next token prediction. To this end, we propose *LazyLLM*, a novel, simple, yet effective technique tailored for speeding up *prefilling*. As depicted in Figure 3, in each generation step, *LazyLLM* selectively computes the KV for tokens important for the next token prediction and "lazily" defers the computation of remaining tokens to later steps when they become relevant. We propose using the attention score of the prior transformer layer to measure the importance of tokens and progressively prune tokens along the depth of the transformer. In contrast to prompt compression works (Li et al., 2023; Jiang et al., 2023; Xu et al., 2023), which permanently reduce the prompt for all the following generation steps, our method allows the model to revive previously pruned tokens, which we found crucial to retain accuracy. Extending progressive token pruning to all generation steps is non-trivial. Specifically, if a token is pruned at generation step $t$, and is revived at generation step $t' > t$, some hidden states would need to be recomputed during step $t'$. To avoid such repetitive computation, we employ an additional caching mechanism, *Aux Cache*, to cache the hidden states of pruned tokens. This enables a computationally efficient pathway to revive pruned tokens, and ensures that the worst runtime of *LazyLLM* is never slower than the baseline.

In summary, the advantages of *LazyLLM* are: (1) **Universal**: *LazyLLM* can be seamlessly integrated with any existing transformer-based LLM to improve inference speed, (2) **Training-free**: *LazyLLM* doesn't require any finetuning and can be directly integrated without any parameter modification, (3) **Effective**: Empirical results on 16 standard datasets across 6 different language tasks shows *LazyLLM* can improve the inference speed of the LLM during both *prefilling* and *decoding* stages.

## 2 RELATED WORK

The increase in the scale of large language models (LLMs) has greatly enhanced their performance but also introduced challenges with respect to their inference efficiency. The inference of generative LLMs consists of two distinct stages as depicted in Figure 1. In particular, extensive computation is needed under long context scenarios to calculate the full KV cache during the *prefilling* stage, resulting in a long time-to-first-token (*TTFT*). This delay causes users to wait several seconds after submitting a prompt before receiving any response from the agent, leading to a poor user experience.

**Efficient Long Context Inference.** Extensive work (Merth et al., 2024; Chen et al., 2023; Beltagy et al., 2020; Kitaev et al., 2020) has been proposed to improve inference efficiency for long context applications by reducing the memory footprint and total computations. Some works have focused on tailoring the architecture of the transformer for long context input. For instance, (Beltagy et al., 2020) introduces a drop-in replacement for standard self-attention and combines local windowed attention with task-motivated global attention. In parallel, Reformer (Kitaev et al., 2020) replaces

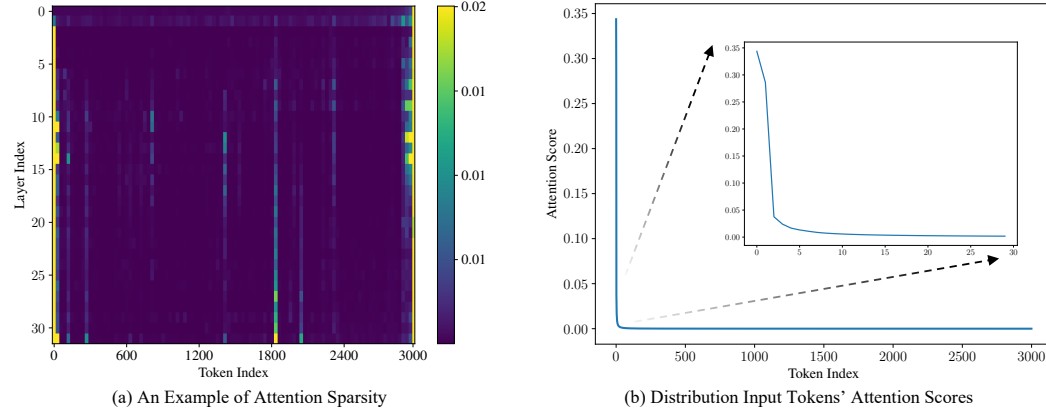

(a) An Example of Attention Sparsity

(b) Distribution Input Tokens' Attention Scores

Figure 2: We visualize the attention scores of input tokens in the prompt w.r.t. to the next token for each layer of Llama 2 7BTouvron et al. (2023). We also plot the distribution of the average attention score across all transformer layers. Result reveals that the attention scores of input tokens w.r.t. to the next token are very sparse, indicating that many tokens in the input prompt are redundant and can be safely removed without affecting the next token prediction.

dot-product attention by one that uses locality-sensitive hashing to reduce its computational complexity. Though the above methods can speed up long context inference, they require significant model architecture change and re-training. This drawback makes them impractical to be applied to existing pre-trained LLMs. Closer to our work are efficient techniques that optimize the KV cache (Zhang et al., 2024; Li et al., 2024; Anagnostidis et al., 2024; Nawrot et al., 2024) by minimizing the KV cache size and data transfer. However, these works only focus on accelerating decoding steps, which are not applicable to reducing *TTFT*.

**Token Pruning.** Previous studies on the sentence classification task (Kim et al., 2022; Anagnostidis et al., 2024; He et al., 2021) has shown that not all tokens (*i.e.* words) in an input sequence are necessary to make a successful prediction. This provides several possibilities for token pruning, which minimizes computational demands by selectively removing less important tokens during inference. For example, (Kim et al., 2022) presents Learned Token Pruning which adaptively removes unimportant tokens as an input sequence passes through transformer layers. In parallel, (He et al., 2021) proposes to reduce width-wise computation via token pruning for transformer-based models such as BERT (Devlin et al., 2018). These aforementioned approaches were designed for tasks requiring only a single iteration of processing, such as text classification. In this work, we extend the idea of token pruning to generative LLMs. Specifically, our method allows the model to dynamically choose different sets of tokens at each generation step, which is crucial to retaining the performance. Furthermore, we also introduce *Aux Cache* to ensure that each token is computed at most once along the whole generation, and ensure the worst runtime of our method is not slower than the baseline.

## 3 *LazyLLM*

### 3.1 BACKGROUND ON LLM INFERENCE

Generative LLM inference consists of two stages: *prefilling* and *decoding* (see Figure 1). In the *prefilling* stage, the model receives the prompt (a sequence of tokens) $\mathcal{T} = \{t_i\}_{i=1}^{N}$ of length N, where $t_i$ denotes a token and $N$ denotes the length of the prompt, then computes and saves the KV cache of each token, and produces the first token $t_{n+1}$. The transformer architecture commonly used in LLMs is a stack of layers where each layer shares the same architecture with a multiple-head self-attention mechanism followed by a multi-layer perception (MLP). The time of *prefilling* is referred to as time-to-first-token (*a.k.a. TTFT*). Following the *prefilling* is the *decoding* steps, where the model appends the generated token $t_{n+1}$ to the input, and subsequently decodes the following token. The *decoding* step is repeatedly performed until the stop criteria are met. While the formula of each decoding step is similar to *prefilling*, the amount of its computation is significantly lower

| | | | Accumulated # of Token Computed |
|---|---|---|---|
| **black**: generated token    **red**: token in computation    **yellow**: retrieved from KV cache    **green**: saved in KV cache but not used    grey: not yet computed | | | |
| **LLM** | Iteration #1 (Prefilling) | LazyLLM is a training free token pruning technique to improve LLM inference with negligible | 13 |
| | Iteration #2 | LazyLLM is a training free token pruning technique to improve LLM inference with negligible performance | 14 |
| | Iteration #3 | LazyLLM is a training free token pruning technique to improve LLM inference with negligible performance loss | 15 |
| **LazyLLM** | Iteration #1 (Prefilling) | LazyLLM is a training free token pruning technique to improve LLM inference with negligible | 4 |
| | Iteration #2 | LazyLLM is a training free token pruning technique to improve LLM inference with negligible performance | 6 |
| | Iteration #3 | LazyLLM is a training free token pruning technique to improve LLM inference with negligible performance loss | 7 |

Figure 3: Comparison between standard LLM and *LazyLLM*. Instead of computing the KV cache of all input tokens at the *prefilling* stage, *LazyLLM* only selectively computes the tokens that are important to the next token prediction, deferring the computation of remaining tokens to later steps. *LazyLLM* significantly optimizes *TTFT* by reducing the amount of computation during *prefilling*. Moreover, as some tokens in the prompt are never selected by *LazyLLM* during the whole generation process (even though theoretically the model *could* use all tokens in the prompt), *LazyLLM* also reduces the total amount of computation and accelerates the overall generation.

thanks to the KV cache. Specifically, with saved KV cache from *prefilling*, all the previous tokens do not need to pass any linear layers in the model.

## 3.2 INFERENCE WITH *LazyLLM*

The overview of the proposed *LazyLLM* framework is illustrated in Figure 4. *LazyLLM* starts with the full context and progressively prunes tokens to gradually reduce the number of computations towards the end of the model. Note, *LazyLLM* allows the model to select different subsets of tokens from the context in different generation steps, even though some of them may be pruned in previous steps. Compared to static pruning which prunes all the tokens at once, dynamic pruning optimizes the next token prediction in each generation step, which is crucial to retaining the performance.

**Progressive Token Pruning.** Prior to this work, token pruning has been successfully applied to optimize LLM inference (Zhang et al., 2024; Li et al., 2024; Adnan et al., 2024; Nawrot et al., 2024). However, these approaches require accumulating the full attention maps of predicting the first few tokens to profile the importance of prompt tokens before starting pruning. Consequently, they are not applicable to reduce *TTFT* as they still require computing all the KV cache at the *prefilling* stage.

In contrast, *LazyLLM* only "lazily" computes the tokens that are important to predict the next token by starting from the *first* iteration of the inference (the *prefilling* step). A key challenge to pruning tokens in the first iteration is determining their importance. Inspired by the early exiting work (Elhoushi et al., 2024) which shows the token hidden states gradually evolve through the transformer layers, we apply layer-wise token pruning in each generation step. Specifically, we use the attention map of the layer $A^l \in \mathcal{R}^{H \times N \times N}$ to determine the importance of input token $t_i$ w.r.t. the next token to be predicted as

$$s_i^l = \frac{1}{H} \sum_{h=1}^{H} A_{h,i,N}^l \tag{1}$$

where $H$ denotes number of attention heads, $N$ is the sequence length, and $A_{h,i,j}$ is the attention probability of the token $t_j$ attending to token $t_i$ at $h^{th}$ head.

After computing the confidence scores of tokens, it is challenging to determine the threshold value to prune the token. Concretely, the threshold can change as the distribution of the attention scores varies between different layers and different tasks. We address this challenge by using the top-$k$ percentile selection strategy to prune tokens. Specifically, token $t_i$ is pruned at layer $l + 1$ if its confidence score $s_i^l$ is smaller than $k^l$th percentile among the input tokens. Once the token is pruned, it is excluded from the computation of all successive layers. In other words, the tokens used in the later layers will be a subset of previous layers.

Our study in Section 5.5 shows the performance changes with different locations of pruning layers and the number of tokens pruned. In particular, when pruning at the same transformer layer, the model's performance gradually decreases as fewer tokens are kept. We also found pruning at later transformer layers consistently has better performance than pruning at earlier layers, suggesting that later layers are less sensitive to token pruning. To achieve a better balance of speedup and accuracy, as shown in Figure 4, we apply progressive pruning that keeps more tokens at earlier transformer layers and gradually reduces the number of tokens towards the end of the transformer.

**Reviving Tokens.** The key difference between *LazyLLM* and previous token pruning work Li et al. (2023); Jiang et al. (2023); Xu et al. (2023); Kim et al. (2022); He et al. (2021) that permanently reduce prompt is *LazyLLM* allows the model to select different subsets of input tokens at each generation step. Since some input tokens pruned at one generation step might become important in subsequent steps, reviving these tokens is crucial for maintaining accuracy. Efficiently reviving pruned tokens during generation is non-trivial. Suppose a token $t_i$ is pruned at one generation step and revived at a later one, a naive implementation to revive $t_i$ requires three steps: 1) updating the keys and values of all previously computed tokens with smaller position IDs than the revived token, 2) computing the keys and values of the revived token, and 3) updating the keys and values of all previously computed tokens with larger position IDs than the revived token. This process leads to multiple updates for the same token, ultimately slowing down generation.

To address this challenge, our implementation skips the first and third steps of updating the keys and values of existing tokens and only computes the revived tokens. Specifically, *LazyLLM* appends the revived tokens to the end of the sequence and uses their position IDs to preserve their original positional information. Consequently, the revived tokens can attend to all tokens selected in previous generation steps, even though these tokens may have later position IDs in the sequence. We found that this implementation is simple yet effective, avoiding repetitive updates of the same tokens and empirically resulting in a negligible performance drop.

**Aux Cache.** In the prefilling stage, there is no KV cache and every token is represented by hidden states. Thus, progressive token pruning can be implemented by removing pruned tokens' hidden states. However, extending the progressive token pruning to the following *decoding* steps is non-trivial. This is because each *decoding* step leverages the KV cache computed in the *prefilling* to compute attention. As the *LazyLLM* performs progressive token pruning at the *prefilling* stage, the KV of tokens pruned at layer $l$ (*e.g.* $T4$ in Figure 4) will not exist in the KV cache of layer $l + 1$. As a reminder, the *LazyLLM* framework allows each generation step to pick a different subset set of tokens from the full input token sequences in every step, regardless of whether they are pruned in previous generation steps or not. For example, during the following *decoding* steps, those pruned tokens (*e.g.* $T4$) that do not exist in the KV cache of layer $l + 1$ may be re-selected to compute attention. In such cases, the model can not retrieve the KV cache of these tokens. An intuitive solution is to pass those tokens again from the beginning of the transformer. However, that would cause repetitive computation for the same token, and eventually slow down the whole generation.

To tackle this challenge, we introduce *Aux Cache* in addition to the original KV cache, which stores the hidden states of those pruned tokens (*e.g.* $T4$ and $T7$ in Figure 4) if their KV is not present in the following layer's KV cache, which could be potentially retrieved for the following iterations. As shown in Figure 4, in each *decoding* step, each transformer layer (*e.g.* layer $l + 1$) first retrieves the KV cache of past tokens if they exist (*e.g.* $T1$ and $T8$). For those tokens that do not exist in the KV cache (*e.g.* $T3$), we could retrieve their hidden states from the *Aux Cache* of its previous layer directly instead of passing through previous layers again. The introduction of *Aux Cache* ensures that each token is computed at most once in every transformer layer, and ensures the worst runtime of *LazyLLM* is not slower than the baseline. It is worth noting that a token resides either in the KV cache or the *Aux Cache*, ensuring that the overall cache size does not exceed that of the baseline.

## 4 IMPLEMENTATIONS DETAILS

We implement *LazyLLM* on Llama 2 (Touvron et al., 2023) and XGen (Nijkamp et al., 2023) and evaluate it on the LongBench (Bai et al., 2023) using HuggingFace[2]. We follow the official GitHub

---

[2]https://github.com/huggingface/transformers/

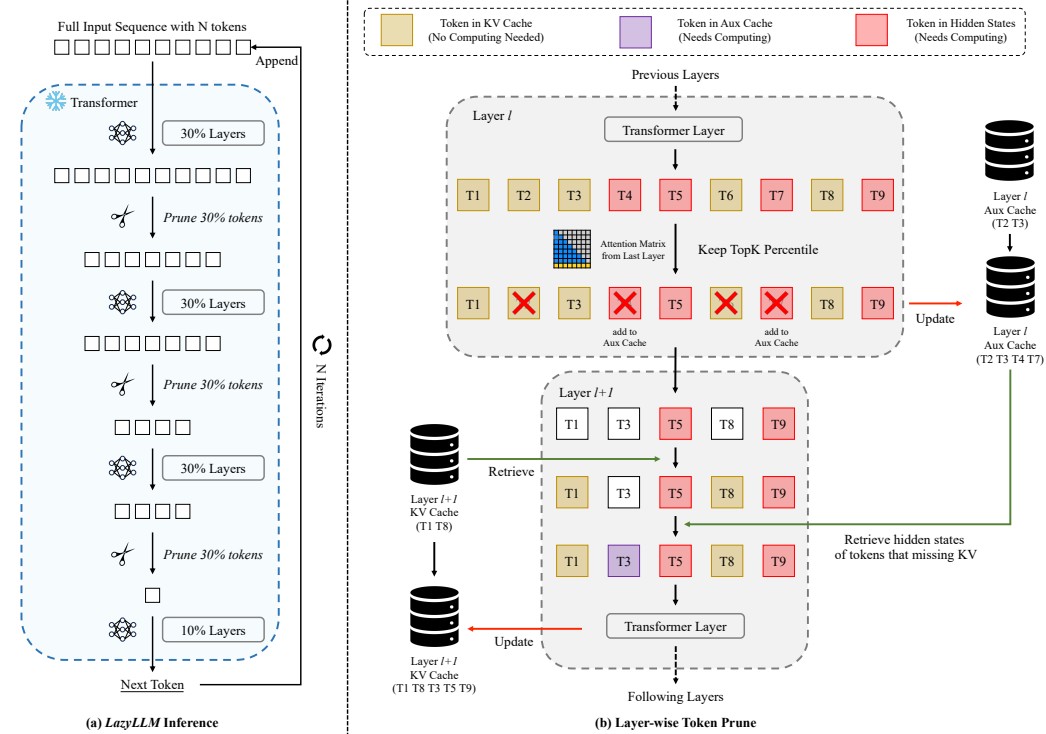

Figure 4: Overview of the *LazyLLM* framework. *LazyLLM* starts with the full context and progressively prunes tokens to gradually reduce the number of computations towards the end of the model. *LazyLLM* allows the model to select different subsets of tokens from the context in different generation steps, which is crucial to retaining the performance.

repository[3] of LongBench for data preprocessing and prompting in all experiments. The LongBench benchmark consists of multiple datasets in different tasks, where each task may have different metrics, including ROUGE-L, F1, Accuracy, and Edit Sim. Following the official evaluation pipeline, we categorize all results over major task categories by computing the macro-average score.

As previously noted, the proposed *LazyLLM* doesn't require any training. Thus, *LazyLLM* uses the exact same existing checkpoints as the baseline, for all models. For inference, we conduct all experiments on NVIDIA A100 GPUs. We measure and report the speedup based on the empirical walltime improvement. Specifically, for *TTFT Speedup*, we measure the empirical walltime between when the prompt is fed to the model, and when the model generates the first token. For *Generation Speedup*, we measure the empirical walltime between when the prompt is fed to the model, and when the model finished generating all output tokens. We add 5 warmup runs for each experiment before starting the time measurement to remove the noise such as loading model parameters.

## 5 EXPERIMENTS

We examine our method using two large language models: Llama 2 7B and XGen 7B. We compare our method with baselines using the same publicly released pretrained checkpoints, without employing any additional training. We perform experiments using LongBench, a multi-task benchmark for long content understanding. The LongBench comprises 16 datasets and covers 6 tasks including single-doc QA, multi-doc QA, summarization, few-shot learning, synthetic tasks, and code completion.

---

[3]https://github.com/THUDM/LongBench

For the metrics, we primarily evaluate the effectiveness and efficiency of each method in the *TTFT* speedup *vs*. accuracy trade-off. Following LongBench, the accuracy (*score*) denotes the macro-averaged scores across datasets in each task. The *TTFT* speedup measures the wall time improvement w.r.t. to the baseline for generating the first token. In analysis, we also assess the impact of our method on % *of Prompt Token Computed* and *Generation* speedup. The % *of Prompt Token Computed* measures the accumulated percent of prompt tokens computed at the end of the generation, which indicates the save of total computation. The *Generation* speedup measures the walltime change w.r.t. to the baseline for completing the entire generation process.

| Tasks | Method | Llama 2 | | XGen | |
|---|---|---|---|---|---|
| | | Score | TTFT Speedup ($\times$) | Score | TTFT Speedup ($\times$) |
| Single-Document QA | Baseline | **25.79** | 1.00 | **25.19** | 1.00 |
| | Random Token Drop | 20.05 | 1.20 | 18.32 | 1.58 |
| | Static Token Pruning | 21.89 | 1.18 | 19.30 | 1.61 |
| | Prompt Compression | 22.88 | 0.12 | 15.31 | 0.20 |
| | *LazyLLM (Ours)* | 25.59 | **1.36** | 25.00 | **1.96** |
| Multi-Document QA | Baseline | **22.43** | 1.00 | **20.71** | 1.00 |
| | Random Token Drop | 16.77 | 1.19 | 14.86 | 1.37 |
| | Static Token Pruning | 19.93 | 2.16 | 17.23 | 2.11 |
| | Prompt Compression | 8.42 | 0.13 | 11.56 | 0.19 |
| | *LazyLLM (Ours)* | 22.31 | **2.34** | 20.68 | **2.65** |
| Summarization | Baseline | 24.65 | 1.00 | **24.85** | 1.00 |
| | Random Token Drop | 24.39 | 1.39 | 24.47 | 1.70 |
| | Static Token Pruning | 24.59 | 1.33 | 24.46 | 1.65 |
| | Prompt Compression | 25.16 | 0.12 | 24.57 | 0.17 |
| | *LazyLLM (Ours)* | **24.75** | **1.46** | 24.74 | **1.91** |
| Few-shot Learning | Baseline | **62.90** | 1.00 | **56.40** | 1.00 |
| | Random Token Drop | 53.93 | 1.19 | 46.35 | 1.62 |
| | Static Token Pruning | 56.54 | 2.16 | 51.93 | 3.17 |
| | Prompt Compression | 24.18 | 0.10 | 23.72 | 0.15 |
| | *LazyLLM (Ours)* | 62.81 | **2.19** | 56.12 | **3.42** |
| Synthetic | Baseline | 4.97 | 1.00 | 5.40 | 1.00 |
| | Random Token Drop | 3.57 | 1.18 | 2.53 | 1.13 |
| | Static Token Pruning | 2.81 | 2.15 | 3.00 | 4.14 |
| | Prompt Compression | 3.20 | 0.12 | 1.42 | 0.17 |
| | *LazyLLM (Ours)* | **4.98** | **2.89** | **5.66** | **4.77** |
| Code Completion | Baseline | **55.18** | 1.00 | **36.49** | 1.00 |
| | Random Token Drop | 44.92 | 1.23 | 32.34 | 1.57 |
| | Static Token Pruning | 37.51 | 1.84 | 32.27 | 2.97 |
| | Prompt Compression | 17.45 | 0.49 | 11.38 | 0.69 |
| | *LazyLLM (Ours)* | 53.30 | **1.94** | 36.47 | **3.47** |

Table 1: Comparisons of *TTFT* speedup *vs*. accuracy on various tasks. Without requiring any training/finetuning, *LazyLLM* consistently achieves better *TTFT* speedup with negligible accuracy drop. Note that the prompt compression approach fails at improving *TTFT* because the overhead of running LLMs to compress the prompt is very computationally expensive.

## 5.1 RESULTS

Table 1 presents the *TTFT* speedup *vs*. accuracy comparisons between *LazyLLM*, standard LLM, and other baselines. In the table, the "baseline" refers to the standard LLM inference. The "random token drop" baseline is based on (Yao et al., 2022) that randomly prunes the prompt tokens before feeding them to the LLMs. We report the average metrics across 5 runs for the "random token drop" baseline. Our "static token pruning" baseline prunes input tokens at once based on their attention score of the first few transformer layers during the *prefilling* stage. We also compare with the prompt compression method (Li et al., 2023) which pruning redundancy in the input context using LLMs. Table 1 shows *LazyLLM* consistently achieves better *TTFT* speedup with negligible accuracy drop across multiple tasks. It is worth noting that the overhead of running LLMs to compress the prompt

is very computationally expensive. Even though the inference on the reduced prompt is faster, the actual *TTFT* of the "prompt compression" baseline is longer than the baseline.

## 5.2 *TTFT* Speedup *vs.* Accuracy

The inference efficiency of *LazyLLM* is controlled using three parameters: 1) the number of pruning layers, 2) the locations of these pruning layers, and 3) the number of tokens pruned within these layers. Increasing the number of pruning layers and pruning more tokens optimize computation by processing fewer tokens, and pruning tokens at earlier layers can save the computations for the successive layers. Prompting these factors will give more overall computation reduction, and offer better *TTFT* speedup. As a side effect, excessively pruning tokens may cause information loss and eventually lead to performance degradation. Similarly, the *TTFT* speedup and accuracy of baselines can vary with different hyperparameters.

We compare *TTFT* speedup *vs.* accuracy in Figure 6 with different hyperparameters. The visualization shows that, without any training, the proposed *LazyLLM* retains the accuracy better than baselines under the same *TTFT* speedup. For example, our method can offer $2.34\times$ *TTFT* speedup in the multi-document question-answering task with negligible ($\leq 1\%$) performance loss. By controlling the pruning parameters, *LazyLLM* provides a good trade-off between accuracy and inference speed as compared to baseline methods. For instance, *LazyLLM* can achieve $3.0\times$ *TTFT* speedup in the multi-document question-answering task with $\leq 10\%$ degradation in accuracy. On the other hand, baseline methods accuracy degrades significantly for similar *TTFT* speed-up. Note that the prompt compression approaches fail at improving *TTFT* because of the compression overhead.

## 5.3 Impact on Overall Generation Speed

To evaluate the impact of the proposed method on the overall generation process, we also profile the *% of Prompt Token Computed* and *Generation* speedup in Table 2. We can find the *% of Token Computed* of *LazyLLM* is less than 100%, indicating that not all tokens in the prompt are selected by *LazyLLM* at the end of the generation, even though theoretically the model *could* use all tokens. Computations in the FFN layers increase linearly, while those in the attention layers grow quadratically with the *% of Token Computed*. A lower *% of Token Computed* indicates *LazyLLM* reduces the total computation, consequently offering additional speedup to the overall generation process across diverse tasks.

## 5.4 Impact on Memory and Computing Cost

By progressively pruning tokens across the transformer layers, *LazyLLM* reduces the size of the attention maps, thereby decreasing the overall memory footprint. Since all tokens are utilized in the initial layers, the peak memory usage remains equivalent to that of the baseline.

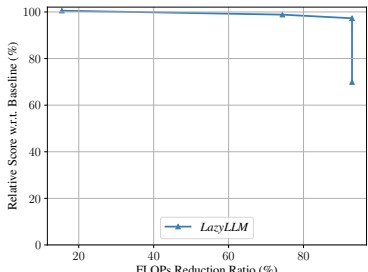

Regarding computational cost, we adopt the methodology from Chen et al. (2024) to calculate the total FLOPs reduction ratio compared to the baseline. Varying the parameters of *LazyLLM* influences both the FLOPs reduction ratio and the model's performance. To illustrate this, we present the FLOPs-performance trade-off curve in Figure 5. The results indicate that *LazyLLM* can significantly lower computational costs with negligible performance drop.

Figure 5: FLOPs-Performance Trade-off Curve of *LazyLLM* for Llama 2 7B evaluated on the Average LongBench Metric.

## 5.5 Drop Rate in Different Layers

In this section, we analyze the effect of the locations of pruning layers, and the number of tokens pruned. In particular, we report a series of experiments using a simplified version of *LazyLLM* that prunes tokens just once within the transformer. For each trial, we position the pruning layer at var-

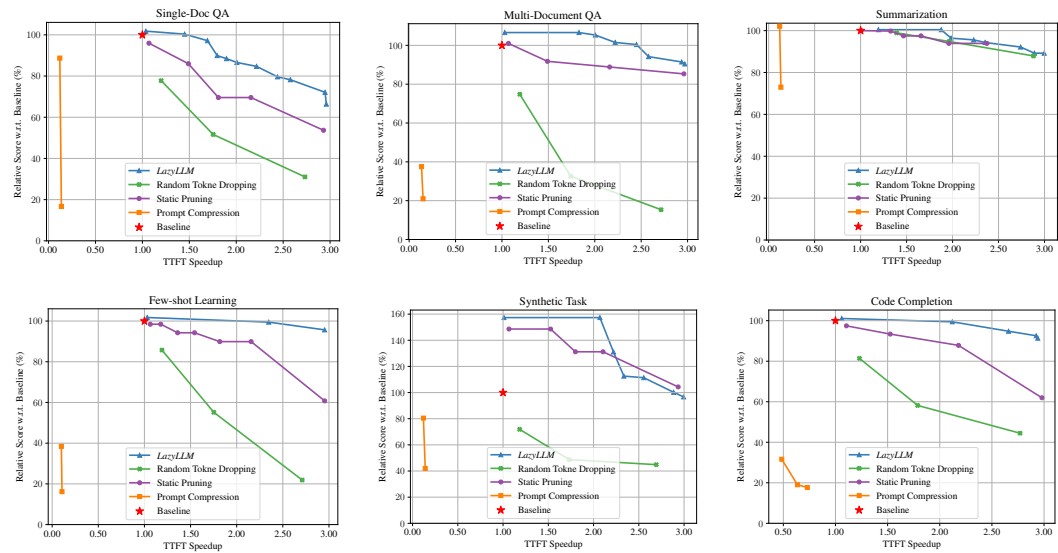

Figure 6: *TTFT* speedup *vs*. accuracy comparison for Llama 2 7B across different tasks.

| TASKS | % OF PROMPT TOKEN COMPUTED | | OVERALL GENERATION SPEEDUP | |
|---|---|---|---|---|
| | LLAMA 2 | XGEN | LLAMA 2 | XGEN |
| SINGLE-DOCUMENT QA | 87.31 | 89.16 | 1.34 | 1.33 |
| MULTI-DOCUMENT QA | 63.94 | 69.60 | 1.56 | 1.70 |
| SUMMARIZATION | 99.59 | 96.11 | 1.02 | 1.09 |
| FEW-SHOT LEARNING | 69.98 | 65.30 | 1.28 | 1.59 |
| SYNTHETIC | 63.73 | 40.54 | 1.79 | 3.16 |
| CODE COMPLETION | 68.57 | 72.61 | 1.01 | 1.16 |

Table 2: The *% of Prompt Token Computed* and *Generation* speedup of *LazyLLM* on various tasks. Reported values are based on the same setting as Table 1. A lower *% of Token Computed* indicates *LazyLLM* reduces the total computation, consequently offering additional speedup to the overall generation process across diverse tasks.

ious levels of the transformer stack and apply different pruning ratios. We perform the experiments for both Llama 2 and XGen, and visualize the results in Figure 7.

The results show both models share a similar trend. As expected, when pruning at the same transformer layer, the model's performance gradually decreases as fewer tokens are kept. Furthermore, pruning at later transformer layers consistently yields better performance compared to pruning at earlier layers, suggesting that later layers are less sensitive to token pruning. Based on these observations, we propose progressive token pruning in Section 3.2, which strategically prunes more tokens in later layers while preserving more in the earlier layers, optimizing the balance between efficiency and performance retention.

## 5.6 PROGRESSIVE KV GROWTH

In this section, we characterize the internals of the model with the token pruning logic. Specifically, we seek to understand what fractions of prompt tokens are cumulatively used and, inversely, not used. This "cumulative token usage" can be equivalently defined as the KV cache size at each given step. Figure 8 presents these cumulative prompt token usage numbers for each of the stages of the LazyLLM.

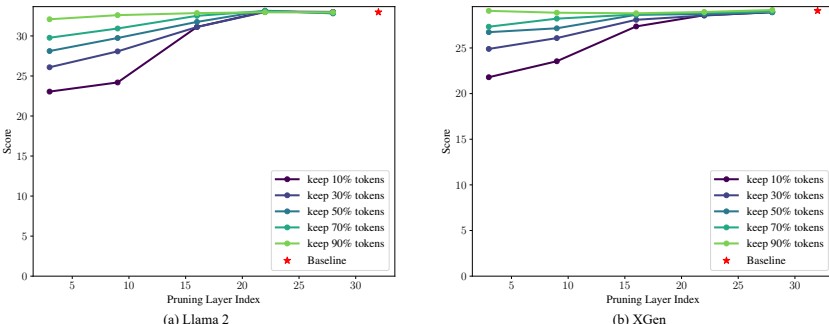

(a) Llama 2                    (b) XGen

Figure 7: Effect of the locations of pruning layers, and the number of tokens pruned. The results of both Llama 2 7B Touvron et al. (2023) and XGen 7B Nijkamp et al. (2023) share a similar trend: 1) when pruning at the same transformer layer, the model's performance gradually decreases as fewer tokens are kept, and 2) Pruning at later transformer layers consistently has better performance than pruning at earlier layers, suggesting that later layers are less sensitive to token pruning.

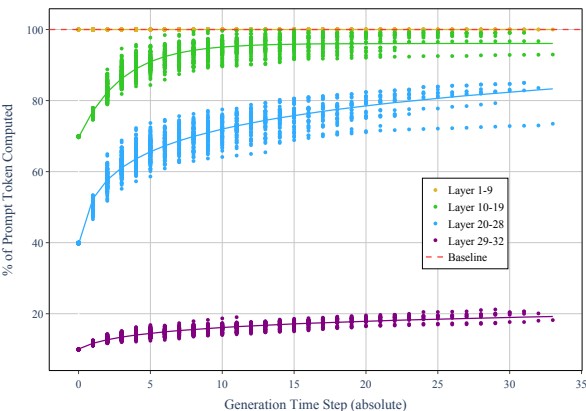

Figure 8: Statistics on number of tokens processed during generation using our *LazyLLM* technique with Llama 2 7B (Touvron et al., 2023). We visualize the statistics of 1000 samples randomly sampled from LongBench. The $x$-axis represents the (absolute) generation time step, and the $y$-axis represents the number of prompt tokens processed at that time step (normalized by the prompt size). We visualize these statistics for various stages within the network. Note that cumulative token usage is upper-bounded by the baseline (evident with early layers).

Our analysis supports the hypothesis that many tokens are never selected by the model (even though theoretically the model *could* use all tokens in the prompt). Since this model retains accuracy on the task(s), we can conclude that the model effectively drops the tokens which do not affect the output quality.

# 6 CONCLUSION

In this work, we proposed a novel *LazyLLM* technique for efficient LLM inference, in particular under long context scenarios. *LazyLLM* selectively computes the KV for tokens important for the next token prediction and "lazily" defers the computation of remaining tokens to later steps, when they become relevant. We carefully examine *LazyLLM* on various tasks, where we observed the proposed method effectively reduces *TTFT* with negligible performance loss. It is worth noting that our method can be seamlessly integrated with existing transformer-based LLMs to improve their inference speed without requiring any fine-tuning.

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

---

**Algorithm 2** Pseudocode of *LazyLLM*

---

**Require:** Input tokens $T = \{t_i\}_{i=1}^N$, transformer model with $L$ layers
**Require:** Pruning thresholds $\{k_l\}_{l=1}^L$ for each layer
 1: Initialize KV Cache and Aux Cache as empty
 2: Initialize active tokens $A_0 \leftarrow T$
 3: Initialize previous attention scores $s_0^i \leftarrow 1$ for all tokens
    {Progressive Token Pruning with Selective Aux Cache Updates}
 4: **for** layer $l = 1$ to $L$ **do**
 5:    **if** $l$ is pruning layer **then**
 6:        Use attention scores $s_{l-1}^i$ from previous layer for pruning decision
 7:        Find $k_l$-th percentile threshold $\theta_l$ of $s_{l-1}^i$
 8:        $A_l \leftarrow \{t_i \in A_{l-1} | s_{l-1}^i \geq \theta_l\}$ {Keep top-k tokens}
 9:
10:        $missing\_tokens \leftarrow \{t_i \in A_l | t_i \notin KV\_Cache \wedge t_i \notin hidden\_states\}$
11:        Retrieve missing tokens from Aux Cache
12:
13:        $pruned\_tokens \leftarrow A_{l-1} \setminus A_l$ {Identify pruned tokens}
14:        $cacheable\_tokens \leftarrow \{t_i \in pruned\_tokens | t_i \notin missing\_tokens \wedge t_i \in hidden\_states\}$
15:        Add $cacheable\_tokens$ to Aux Cache
16:    **else**
17:        $A_l \leftarrow A_{l-1}$
18:    **end if**
19:    Compute layer outputs for tokens in $A_l$
20:    Compute current layer attention scores $s_l^i$ using Eq.(1) for tokens in $A_l$
21:    Update KV Cache for tokens in $A_l$
22: **end for**
23: **return** Final hidden states for tokens in $A_L$

---

# A  APPENDIX

## A.1  PSEUDOCODE

Algorithm 1 presents presents *LazyLLM*'s progressive token pruning strategy enhanced with an auxiliary caching mechanism. For each transformer layer, the algorithm first uses attention scores from the previous layer to make pruning decisions, maintaining only the most relevant tokens. After pruning, it identifies tokens missing from both KV Cache and hidden states, retrieving them from the Auxiliary Cache when needed.

# B  VISUAL EXAMPLE

To better illustrate how *LazyLLM* operates, we present a detailed walkthrough of our method in Figure 9. Consider a simple example where the model processes the input "*LazyLLM is a training free token pruning technique to improve LLM inference with*" and generates subsequent tokens "*negligible performance loss*". The visualization demonstrates how *LazyLLM* evolves through different stages of generation.

During the prefilling stage, instead of computing all tokens in the prompt, methodname selectively processes only those tokens deemed important for the next token prediction. In our example, methodname initially processes only 13 tokens compared to the baseline's full sequence processing. Notably, when generating the first token "negligible", methodname focuses on key contextual tokens like "LazyLLM", "improve", and "inference", while deferring the computation of less relevant tokens.

In subsequent decoding steps (Step #2 and Step #3), methodname continues to operate efficiently by:

1. Reusing previously computed KV cache values when possible

2. Selectively computing only newly important tokens that were deferred earlier

3. Maintaining the ability to revive previously pruned tokens if they become relevant

This dynamic approach results in significantly reduced computation, compared to the baseline which processes all tokens at every step. The visualization clearly shows how tokens in red indicate active computation, and green denotes retrieved from KV cache.

This example demonstrates how *LazyLLM* achieves substantial computational savings without sacrificing model performance. The method's ability to dynamically adjust token selection at each generation step, while maintaining efficiency through strategic caching, represents a key advancement over static pruning approaches.

# Method - Example Explained
## Prefilling Stage

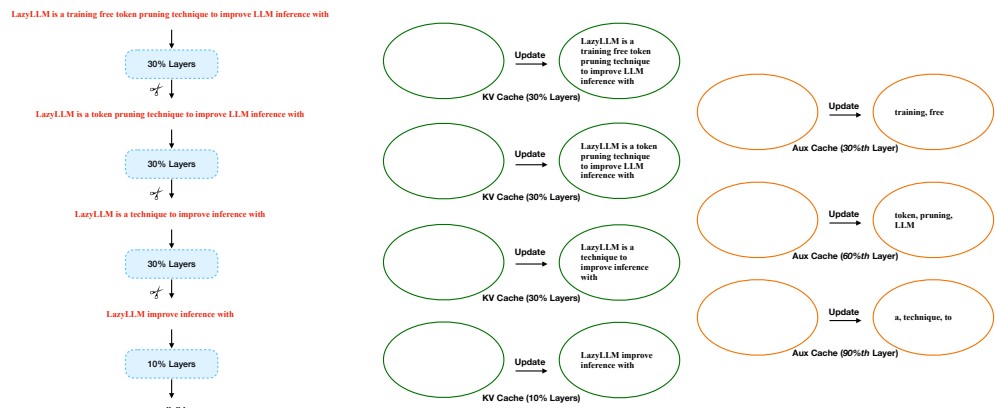

# Method - Example Explained
## Generation Stage - Step 1

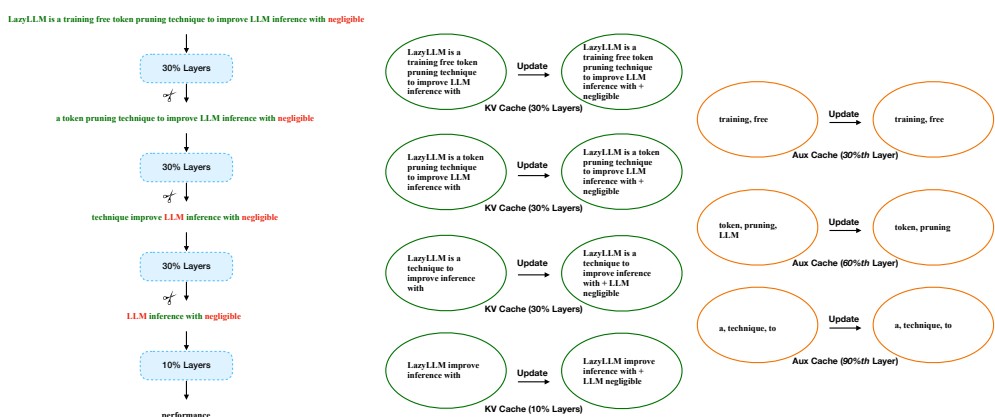

# Method - Example Explained
## Generation Stage - Step 2

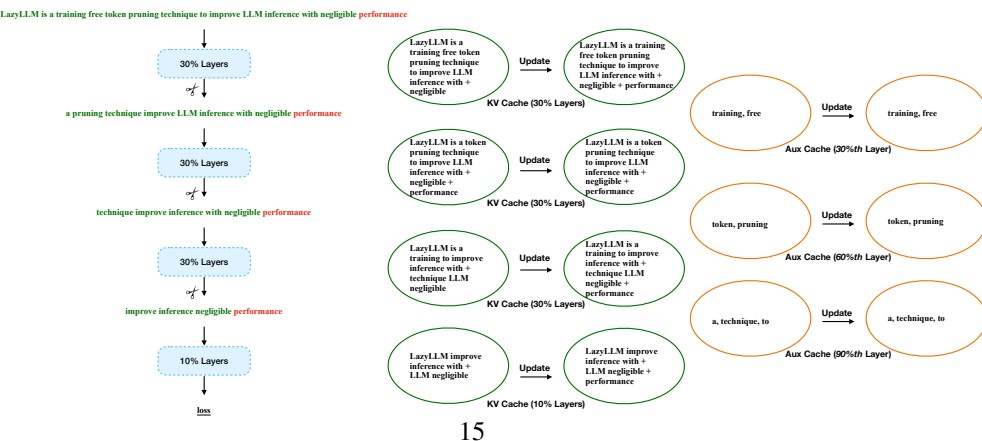

Figure 9: Visual Example