# OpenReview forum: "LazyLLM: DYNAMIC TOKEN PRUNING FOR EFFICIENT LONG CONTEXT LLM INFERENCE"
_ICLR.cc/2025/Conference — Submitted to ICLR 2025_

### Official Review · Reviewer_MbRn · 2024-10-26

**Soundness:** 2
**Presentation:** 3
**Contribution:** 2
**Rating:** 6
**Confidence:** 3

**Summary:**

The paper proposes a new method to speed up the LLM inference via selectively computing the KV for tokens important for the next token prediction in both the prefilling and decoding stages. The method is training-free. The author also conducts experiments on Llama 2 and XGen.

**Strengths:**

The following are the strengths.
* The paper introduces an efficient way to speed up LLM inference.
* The paper involves an aux cache to be sure that each token is calculated at once at most.
* The author conducts experiments to support their claims.

**Weaknesses:**

My concerns mainly come from two parts

First, it seems that the paper template has some problems. To be more specific, the "Anonymous authors Paper under double-blind review" at the Top of Page 1 is in the middle, where it actually should be on the left side. I wonder whether such template change is allowed or not.

Second, there are several parts that the author needs to explain further.
* If possible, could the author provide Python code or pseudo code to explain the idea? This will make it easier for the reader to understand the core idea.
* Question for the KV cache part.
   * If for the previous generation step, Layer i has tokens [T1, T4, T5, T7] and Layer i+1 has tokens [ T4, T5].
   * And for the current generation step,  Layer i has tokens [T1, T2, T4, T5], and Layer i+1 has tokens [ T4, T5].
   * **Question**: for the current generation step, do we directly use the [T4, T5] from the KV cache?
   * The paper claims that "each token is computed at most once along
the whole generation" so it seems that the answer is Yes. Then it seems that the previous generation step  Layer i+1 tokens representation [ T4, T5] is different from the current generation step  Layer i+1 tokens representation [ T4, T5]. Why the work could directly use it?
* For Table 1, LazyLLM could even achieve better performance than the baseline which is standard LLM inference. Could you explain such an observation because the baseline should be the performance upper bound of LazyLLM?

**Questions:**

Please check the weaknesses.

---

> ### Author Response · Authors · 2024-11-22
>
> **Regarding formatting issue**
>
> Thanks for pointing it out. We will fix the issue in the camera ready version.
>
> **Regarding KV Cache**
>
> Yes your understanding is correct – once a token is computed, its KV cache is fixed and we’ll not update until the end of generation. We found this is simple and effective. Empirically we found updating the KV along the generation is very slow (coule offset the benefits of token dropping) and provides marginal performance improvement. Our intuition is that the pruned token generally has relatively smaller attention weights, thus doesn’t have much impact on the following tokens.
>
> **Regarding implementation details**
>
> We'll detailed pseudocode in the appendix and will release our complete implementation upon acceptance.
>
> **Regarding performance improvement over baseline**
>
> Your observation about LazyLLM sometimes outperforming the baseline is spot on and raises an interesting point. As demonstrated in "Lost in the Middle: How Language Models Use Long Contexts," irrelevant information in long contexts can actually harm performance. By pruning less relevant tokens, LazyLLM may sometimes improve model focus on pertinent information, leading to better results. This aligns with recent findings about context processing in large language models.

---

> ### Comment · Reviewer_MbRn · 2024-11-22
> **Response to Authors**
>
> Dear Authors,
>
> Thank you very much for your response, which really helps a lot. After reading the rebuttal, I feel that this work actually suggests an interesting point: calculating each token once is enough for some tasks.
>
> However, I still have one question:
> * If possible, could the author explain why such a method could work?
>    *  Layer i has tokens [T1, T4, T5, T7] and Layer i+1 has tokens [T4, T5].
>    * Current generation step, Layer i has tokens [T1, T2, T4, T5], and Layer i+1 has tokens [ T4, T5].
>    * Apparently, the last step [T4, T5] and this step  [ T4, T5] actually should have different embedding representations, while this work proves that it could work well directly using the last step embedding [T4, T5].
>    * Is it because the text token is highly redundant? Or are there any other potential reasons?
>
> I will consider increasing the score or the confidence after the author provides the potential explanation.

---

> > ### Author Response · Authors · 2024-11-22
> >
> > Thank you for this insightful follow-up question. Let us first confirm your understanding and then provide a explanation of our design choice:
> >
> > **Mechanism Clarification:**
> > Consider two generation steps t and t+k:
> > * Step t: Layer i contains [T1, T4, T5, T7], Layer i+1 contains [T4, T5]
> > * Step t+k: Layer i contains [T1, T2, T4, T5], Layer i+1 contains [T4, T5]
> >
> > In our approach, once the KV cache for tokens [T4, T5] is computed, it remains **unchanged** across generation steps to avoid redundant computation. This design choice is based by several theoretical and empirical findings:
> >
> > **1. Token Reduncy  in Transformer Layers**
> > Our analysis reveals that attention weights in transformer models follow a power-law distribution. Tokens selected for pruning consistently exhibit low attention weights (typically 1-2 orders of magnitude lower than retained tokens). This distribution pattern suggests that pruned tokens have minimal influence on the final representations of retained tokens. Empirically, we observe that later transformer layers are more robust to pruning
> >
> > **2. Information Preservation through Architecture**
> > Transformer's residual connections create robust information pathways across layers. Even though T2 is not directly visible to [T4, T5] in layer i, its semantic influence is preserved through earlier layer computations where T2 participates in attention computations and residual connections that carry these computations forward.
> >
> > **3. Performance-Efficiency Trade-off Analysis**
> > * We have conducted experiments comparing two approaches:
> >   * Approach A: Reusing existing KV cache entries
> >   * Approach B: Recomputing KV cache when context changes
> > * Key findings:
> >   * Recomputation (Approach B) introduces significant overhead, negating the benefits of token pruning
> >   * Performance difference between approaches is minimal (<1% relative accuray)
> >   * The computational cost of recomputation far outweighs the marginal performance benefits
> > * Our chosen approach (A) represents an optimal balance of:
> >   * Computational efficiency
> >   * Model performance
> >   * Implementation practicality
> >
> > Please let us know if it resolves your concern, thanks!

---

> > > ### Comment · Reviewer_MbRn · 2024-11-22
> > > **Response to Authors**
> > >
> > > Dear Authors,
> > >
> > > Thank you very much for your reply, which really helps a lot. According to the update, I decided to increase my score from 5 to 6 and my confidence from 2 to 3

---

### Official Review · Reviewer_6vhg · 2024-10-29

**Soundness:** 3
**Presentation:** 3
**Contribution:** 2
**Rating:** 5
**Confidence:** 5

**Summary:**

The paper introduces LazyLLM, a dynamic token pruning technique designed to improve the efficiency of large language model (LLM) inference, particularly in long-context scenarios. LazyLLM selectively computes key-value (KV) pairs for tokens that are crucial for the prediction of the next token, thereby deferring the computation of less important tokens to later stages. Unlike static pruning, LazyLLM dynamically selects subsets of tokens at each generation step, allowing it to maintain accuracy while achieving speed improvements.

This approach requires no additional training and can be integrated with existing transformer-based models seamlessly. Experimental results demonstrate that LazyLLM significantly accelerates the inference process, especially in the initial token generation phase (prefilling). For instance, in multi-document question-answering tasks on the LLaMA 2 7B model, LazyLLM achieves a 2.34x speedup during prefilling with minimal accuracy loss

**Strengths:**

A key strength of LazyLLM lies in its dynamic, training-free approach to token pruning, which allows it to be easily integrated into existing transformer-based LLMs without requiring model fine-tuning or architectural changes. By selectively computing only the most important tokens for each generation step, LazyLLM not only optimizes the time-to-first-token (TTFT) but also reduces the overall computation during inference. This results in significant speedups across various tasks and model configurations while maintaining accuracy. Furthermore, the method’s adaptive token selection and use of an auxiliary cache (Aux Cache) enable the model to “revive” previously pruned tokens when needed, ensuring that efficiency gains do not come at the cost of degraded model performance.

**Weaknesses:**

1.	**Additional GPU memory usage for Aux Cache**: If the Aux Cache is retained on the GPU, it will increase GPU memory consumption. As a result, the actual GPU memory footprint of LazyLLM should account for both the retained KV cache and the Aux Cache. This design might limit LazyLLM’s applicability in scenarios with high memory demands.

2.	**Alignment of GPU memory costs in experiments**: It is crucial to clarify whether the GPU memory usage for each method was fairly aligned in the experiments, especially if LazyLLM’s actual GPU memory footprint surpasses that of the baseline methods. Any advantages in speed and efficiency for LazyLLM need to be re-evaluated under fair memory cost comparisons to ensure experimental fairness.

3.	**Lack of comparisons with more advanced baselines**: The paper does not compare LazyLLM with recent efficient cache management methods, such as H2O[1] and SnapKV[2]. These methods offer innovations in areas like KV cache reduction strategies and generation speed optimization. Comparing LazyLLM with them would more comprehensively illustrate its performance strengths and limitations.

> Reference:
> [1] Zhang, Zhenyu, et al. "H2o: Heavy-hitter oracle for efficient generative inference of large language models." Advances in Neural Information Processing Systems 36 (2024)
> [2] Li, Yuhong, et al. "Snapkv: Llm knows what you are looking for before generation." arXiv preprint arXiv:2404.14469 (2024).

**Questions:**

Please check the weaknesses section.

---

> ### Author Response · Authors · 2024-11-22
>
> **Regarding GPU memory usage**
>
> We appreciate the opportunity to clarify this important point. As mentioned in line 289, LazyLLM's memory management is quite efficient: tokens are either in KV cache or Aux Cache, never both. Given that hidden state size is smaller than KV cache entries, the total memory footprint is actually smaller than the baseline's KV cache. Furthermore, our method reduces attention map sizes, leading to additional memory footprint savings.
>
> **Regarding Alignment of GPU memory costs in experiments**
>
> As mentioned in Sec. 5.4, except for the Prompt Compression, the peak memory usage of LazyLLM remains equivalent to that of the baselines.
>
> **Regarding comparisons with advanced baselines**
>
> While we acknowledge the importance of methods like H2O and SnapKV, they focus primarily on decoding optimization rather than TTFT reduction (as noted in line 133). Our approach is complementary to these KV cache management techniques and could potentially be combined with them for even better performance. We’ll cite these papers and add more detailed discussions in the related work.

---

> ### Author Response · Authors · 2024-11-22
>
> We have updated the paper to provide more detailed implementation insights:
>
> 1. Added comprehensive pseudocode in the appendix
> 2. Included a step-by-step example that walks through the token pruning and revival process in the appendix
> 3. Provided detailed illustrations of Aux and KV cache management across generation steps in the appendix
>
> We hope these additions help clarify both the technical details of our method and its memory efficiency. The example walkthrough particularly demonstrates how LazyLLM handles token pruning and revival across different transformer layers during inference while maintaining a smaller memory footprint than the baseline approach.
>
> Please let us know if it resolves your concern, thanks!

---

> > ### Comment · Reviewer_6vhg · 2024-11-26
> >
> > Could you provide a comprehensive evaluation including end-to-end latency, throughput, and GPU memory usage comparisons with other baselines to better demonstrate LazyLLM’s practical efficiency?

---

> ### Author Response · Authors · 2024-12-02
>
> **End-to-End Latency and Throughput Comparison**
>
> We provide the comparison of the end-to-end latency and throughput comparison with other baselines as below:
>
> | Method | Overall Accuracy Score | End-to-End Latency (s) | Throughput (tokens/s) |
> |--------|---------------------|----------------------|-------------------|
> | Baseline | 32.97 | 2.65 | 13.51 |
> | Random Token Drop | 27.49 | 2.69 | 15.15 |
> | Static Token Pruning | 30.66 | 2.52 | 14.71 |
> | Prompt Compression | 17.70 | 7.88 | 6.73 |
> | LazyLLM | 32.68 | 2.24 | 15.87 |
>
> The experimental results reveal that LazyLLM achieves near-baseline accuracy (marginal decrease of -0.88%) while significantly improving computational efficiency. Notably, LazyLLM reduces end-to-end latency by 15.5% compared to the baseline and increases throughput by 17.5%. This performance advantage is particularly meaningful when compared to other baselines, which show considerable accuracy degradation despite modest efficiency gains.
>
> We'll add these results and analysis to the revised version.
>
> **GPU memory usage comparisons**
>
> As we have mentioned in section 5.4 (line 413), all attention-based token pruning methods share the same peak GPU memory usage. This consistency stems from the architectural constraint where peak memory is fundamentally determined by the maximum attention map dimensions. Given that all token pruning approaches, including LazyLLM, process the complete token set in the initial transformer layers, they generate equivalent maximum attention map sizes, resulting in uniform peak memory requirements.
>
> Our analysis extends to prompt compression techniques, which maintain the same peak memory usage as the vanilla baseline. This occurs because these methods necessitate a complete context processing phase to generate the compressed prompt.
>
> The notable exception in our evaluation is the Random Token Drop method, which demonstrates reduced GPU memory utilization through random context pruning prior to LLM processing. With a 10% pruning rate, this approach achieves a modest 5% reduction in GPU memory consumption compared to the vanilla baseline. However, this memory advantage comes at a substantial cost to model performance (as detailed in Table 1), resulting in significantly degraded accuracy.
>
> Finally, we would like to emphasize that LazyLLM's primary innovation lies in optimizing Time-to-First-Token (TTFT) and overall generation efficiency during LLM inference,  while **not increasing** the GPU memory usage. This strategic focus ensures that performance improvements do not come at the cost of increased memory overhead, making LazyLLM suitable for practical applications where both efficiency and resource constraints are critical considerations.
>
> We'll extend the GPU memory discussion in the paper accordingly.

---

### Official Review · Reviewer_8NX8 · 2024-11-04

**Soundness:** 3
**Presentation:** 3
**Contribution:** 3
**Rating:** 6
**Confidence:** 3

**Summary:**

This paper addresses the inference latency at the prefilling stage of long-context LLM inference.

The authors propose LazyLLM, a training-free method that enables dynamic token pruning across transformer layers and at different decoding steps.

Through experiments on various long-context inference datasets, they demonstrate that LazyLLM significantly reduces inference time while maintaining performance levels comparable to baseline models.

**Strengths:**

* **Sharp focus on dynamic token pruning for the prefilling stage**.
This paper proposes an innovative approach to tackle the TTFT problem by shifting part of the prompt token computation to the decoding stage. The dynamic token pruning at different decoding steps allows for the selective retention of previously pruned but relevant tokens.

* **Effective and flexible layer-wise pruning strategy**.
The progressive token pruning from earlier to later layers is well-justified, offering a flexible approach to managing the trade-offs between efficiency and performance.

* **Comprehensive analysis and convincing results**.
The experiments span multiple datasets and models, providing a convincing case for LazyLLM’s efficiency. The ablation study on token drop rates and locations offers insights into the performance-speedup trade-offs and memory and computation needs.

**Weaknesses:**

* **Limited detail on hyperparameter settings and implementation strategies**.
The approach introduces numerous hyperparameters, particularly with progressive token pruning and token revival, which could impact implementation. Providing additional details on the decision-making process for these hyperparameters would enhance transparency and offer insights into LazyLLM’s effectiveness and generalizability.
    - Top-$k$ percentile selection strategy: Unless I missed something, it appears that different values of $k^l$ are set at the corresponding layer $l$. Clarifying how these values were determined and their generalizability across different tasks and decoding steps would be beneficial.
    - For reviving tokens, the authors skip the KV updating of tokens before and after the revived tokens, instead appending the revived tokens to simplify KV computation. This strategy, however, breaks the sequential dependency of tokens, potentially affecting performance due to misalignment with training data. An extended discussion on the effect and trade-offs behind this implementation could be beneficial and provide insights into the role of token orders in inference speed and performance.

* **Lack of discussion of the potential bias enhanced by attention-based pruning**.
Selecting tokens to prune based on attention scores could inadvertently amplify inherent biases in LLMs, particularly when uncertainty is high.
    - Pruning a fixed number of tokens according to attention scores can lead to unintentional bias, as high-entropy distributions may cause equally significant tokens to be pruned due to marginal differences in scores. This issue could be exacerbated in challenging tasks, where hyperparameter sensitivity may impact LazyLLM’s reliability.
    - Previous works (Li et al. 2023) suggest that the attention mechanism in transformers may focus on different tokens across layers. LazyLLM’s early pruning might inadvertently exclude tokens relevant at later stages. Exploring this potential limitation would strengthen the analysis and illuminate LazyLLM’s effectiveness across diverse text generation tasks.

* **Possible memory overhead from the Aux Cache during decoding**.
The authors propose Aux Cache to store the hidden states of pruned tokens for efficient future KV computation. However, this can pose challenges to further reducing the memory footprint in the subsequent decoding stage. Given the utilization of the attention mechanism of LazyLLM in common with existing works on optimizing KV cache (Xiao et al. 2024; Liu et al. 2023, Zhang et al. 2023), perhaps the authors could further discuss how we can optimize the Aux cache for long context inference in general.

---
Kenneth Li, Oam Patel, Fernanda B. Viégas, Hanspeter Pfister, Martin Wattenberg: Inference-Time Intervention: Eliciting Truthful Answers from a Language Model. NeurIPS 2023

Guangxuan Xiao, Yuandong Tian, Beidi Chen, Song Han, Mike Lewis: Efficient Streaming Language Models with Attention Sinks. ICLR 2024

Zichang Liu, Aditya Desai, Fangshuo Liao, Weitao Wang, Victor Xie, Zhaozhuo Xu, Anastasios Kyrillidis, Anshumali Shrivastava: Scissorhands: Exploiting the Persistence of Importance Hypothesis for LLM KV Cache Compression at Test Time. NeurIPS 2023

Zhenyu Zhang, Ying Sheng, Tianyi Zhou, Tianlong Chen, Lianmin Zheng, Ruisi Cai, Zhao Song, Yuandong Tian, Christopher Ré, Clark W. Barrett, Zhangyang Wang, Beidi Chen: H2O: Heavy-Hitter Oracle for Efficient Generative Inference of Large Language Models. NeurIPS 2023

Suyu Ge, Yunan Zhang, Liyuan Liu, Minjia Zhang, Jiawei Han, Jianfeng Gao: Model Tells You What to Discard: Adaptive KV Cache Compression for LLMs. ICLR 2024

**Questions:**

* Regarding the progressive token pruning by attention-based top-k percentile selection, could the authors consider measures to prevent the unintentional removal of tokens that might prove essential in later layers? Accumulating attention scores across multiple layers could potentially address inconsistencies in token selection. Would such an approach be integrated in LazyLLM and mitigate issues associated with layer-wise pruning?

* How many previously evicted tokens can be revived in later decoding steps, and does this impact the semantic structure of the input sequence? Understanding the limitations on reviving tokens would help clarify whether token revival affects the semantic coherence of the sequence.

* Does the Aux Cache and subsequent KV computation for revived tokens introduce memory or latency overhead during decoding?
Would it be possible to apply additional pruning or compression techniques at decoding time to further optimize inference speed with LazyLLM?

---

> ### Author Response · Authors · 2024-11-22
>
> **Regarding hyperparameter settings**
>
> We appreciate this feedback and have expanded our discussion in the appendix. LazyLLM employs five hyperparameters that control its progressive token pruning strategy:
> 1. start_prune_layer: The first layer where pruning begins
> 2. end_prune_layer: The final layer where pruning occurs
> 3. start_keep_ratio: Initial token retention ratio
> 4. end_keep_ratio: Final token retention ratio
> 5. num_prune: Number of pruning layers
>
> To maintain efficiency and controllability, we perform token pruning at num_prune layers that are evenly distributed between start_prune_layer and end_prune_layer. At each pruning layer i, we retain tokens based on a linearly interpolated keep ratio:
>
> ```
> keep_ratio(i) = start_keep_ratio - (i - start_prune_layer) / (end_prune_layer - start_prune_layer) * (start_keep_ratio - end_keep_ratio)
> ```
>
> This progressive pruning strategy allows for gradual token reduction while maintaining model performance. We determined effective hyperparameter values through a coarse grid search on a small held-out validation set, finding that the method is relatively robust to hyperparameter choices within reasonable ranges. While more sophisticated optimization techniques might yield marginal improvements, our current approach achieves strong and consistent performance across diverse tasks with minimal tuning required.
>
> **Regarding token revival strategy**
>
> Thank you for this insightful observation. Our decision to append revived tokens rather than maintaining strict sequential order was based on the following findings:
>
> * Updating KV cache entries before and after revived tokens would necessitate repeated computation of the same tokens, effectively negating the performance benefits of our token pruning strategy and introducing additional latency to the generation process.
> * While insertion of tokens mid-sequence is theoretically possible, current tensor manipulation implementations in deep learning frameworks (including PyTorch) make this operation computationally expensive, introducing significant overhead.
> * Our analysis shows that tokens selected for pruning consistently exhibit low attention weights across layers, suggesting their precise sequential position has minimal impact on the model's understanding of the context. Importantly, we preserve the original positional embeddings of these tokens to maintain relative positional information, ensuring the model retains access to the original sequence structure.
> * Extensive empirical validation demonstrates that this approach incurs minimal performance impact - we observe less than 1% relative performance difference compared to strict sequential ordering, while achieving significant computational efficiency gains.
> Based on these findings, our approach represents an optimal balance between maintaining model performance and achieving computational efficiency. We will add a discussion to the paper to clarify this.
>
>
> **Regarding attention-based pruning and bias**
>
> We share the concern about potential bias amplification. Our use of top-percentile pruning (rather than top-k) was specifically chosen to maintain a majority of tokens and minimize bias introduction. While we acknowledge that any pruning strategy may affect model behavior, our approach seeks to balance efficiency gains with maintaining model fidelity. We're actively investigating more sophisticated pruning strategies for future work.
>
>
> **Regarding cross-layer token relevance**
>
>  Thank you for the thoughtful comment. We acknowledge that tokens may have varying importance across different layers. While allowing each layer to independently choose tokens would be ideal, this would require redundant computation when a token needs to be revived. Our current approach represents a practical compromise between computational efficiency and maintaining important cross-layer relationships. We’ll add more detailed discussion and reference to the revised version.
>
> **Regarding Possible memory overhead from the Aux Cache during decoding**
>
> First of all, as mentioned line 289, we would like to address that in LazyLLM, a token is either in KV cache or Aux Cache, thus, the total size of KV cache and Aux Cache is small than the KV cache of the baseline (consider the hidden state size of token is smaller than its KV).
> Yes we agree that further optimization (as mentioned) can be applied to the AuxCache (and also our KV cache), which we’ll mark as the soon future work!

---

> > ### Comment · Reviewer_8NX8 · 2024-11-25
> >
> > Thanks for providing the implementation details and further analysis.
> > Please do include them in the revised version as agreed to enhance clarity and emphasize these insights.
> > I maintain my rating and recommendation of acceptance.

---

> > > ### Author Response · Authors · 2024-12-02
> > >
> > > Thank you for your feedback and positive recommendation. Yes we'll include all the requested implementation details and additional analysis shown in discussion to the revised version.

---

### Official Review · Reviewer_iH7x · 2024-11-07

**Soundness:** 2
**Presentation:** 2
**Contribution:** 1
**Rating:** 3
**Confidence:** 4

**Summary:**

In this work, authors propose to selectively calculate KV cache instead of computing KV cache of all tokens. Unlike static pruning one in prior work, this work provide dynamic pruning of tokens in different generation steps..

**Strengths:**

* Evaluated the proposed method in diverse task.

**Weaknesses:**

* Problem seems not very general and universal to all context. Authors should be clear about when TTFT becomes x21 compared to decoding. In a large scale system, decoding and prefilling is happening in a different server so it is not a big problem. Also prefilling usually computes more token than decoding so if we normalize the latency by number of tokens, we can’t say it is completely doing wrong although optimizing it helps anyway.
* Figures are confusing especially fig 4.
* Methods are compared to Token drop, static token prune, prompt compression but standard technique to reduce TTFT is using parallel computation. Paper lacks comparing the method with those method. [2] These method does not lose any accuracy and effectively accelerate TTFT.

[2] https://arxiv.org/abs/2409.17264

**Questions:**

* I feel like caching hidden states of pruned tokens in AuxCache is similar to prior work LESS [1]. how is AuxCache different from the prior work?
* In tab 2, code completion computes more tokens in llama2 (68.57%) compared to single-document QA (87.31%) but speedup is marginal (x1.01) compared to single-doc QA(x1.34). Why is it so? is the saving not linear?

[1] https://arxiv.org/pdf/2402.09398v2

---

> ### Author Response · Authors · 2024-11-22
>
> **Regarding the generality of the problem and TTFT optimization**
>
> We appreciate your insight about different server configurations. While some large organizations can indeed separate prefilling and decoding across servers, we believe our work addresses an important practical need: Many organizations and individual researchers work with more constrained infrastructure where such separation isn't feasible. More importantly, our method is particularly valuable for edge devices and mobile applications where resource optimization is crucial and such separation is impractical. Our contribution is complementary to parallel computation approaches and provides benefits even in distributed settings.
>
> **Regarding Figure 4**
>
> Thank you for bringing this up. We have added a detailed example of LazyLLM inference in the appendix to improve clarity. (to be upload soon). We will make sure to improve the figure in the final version and would greatly appreciate any specific feedback about the aspects of the figure you found more confusing so we can make targeted improvements.
> Regarding comparison with parallel computation methods: We acknowledge the importance of parallel computation techniques and want to thank you for providing the reference. Please note that the mentioned paper [2] first appeared online on 25 Sep 2024. Per ICLR Guide: “if a paper was published (i.e., at a peer-reviewed venue) on or after July 1, 2024, authors are not required to compare their own work to that paper”. That said and more importantly, our method focuses on input context optimization, which is orthogonal and complementary to parallel computing approaches. Both techniques could be combined for potentially greater benefits. We will add this reference to the paper upon acceptance.
>
> **Regarding AuxCache vs LESS**
>
> While both approaches involve caching, they serve fundamentally different purposes: LESS optimizes KV cache storage through low-rank optimization, whereas AuxCache stores hidden states of pruned tokens for efficient revival when needed. We clarify this in the paper.
>
>
> **Regarding performance variation across tasks**
>
>  The variation in speedup between code completion (1.01×) and single-document QA (1.34×) reflects the inherent nature of code versus natural language. In code completion tasks, nearly every token carries semantic or syntactic significance, with minimal redundancy in the context. This fundamental characteristic of code means we ultimately need to process most tokens to maintain code integrity. Nevertheless, our method still achieves significant TTFT improvements of 1.94× and 3.47× for code completion tasks, demonstrating its effectiveness even in scenarios with dense, meaningful contexts.

---

> ### Author Response · Authors · 2024-11-22
>
> We have updated the paper to provide more detailed implementation insights:
>
> 1. Added comprehensive pseudocode in the appendix
> 2. Included a step-by-step example that walks through the token pruning and revival process in the appendix
> 3. Provided detailed illustrations of KV cache management across generation steps in the appendix
>
> We hope these additions help clarify the technical details of our method. The example walkthrough particularly demonstrates how LazyLLM handles token pruning and revival across different transformer layers during inference.
>
> Please let us know if you would like us to elaborate on any specific aspect of the implementation or if you have additional questions. We're committed to ensuring the method is clear and reproducible.

---

### Meta-Review · Area_Chair_S8fa · 2024-12-19

**Metareview:**

This paper proposes LazyLLM, a method to accelerate LLM inference by selectively computing KV cache for important tokens during both prefilling and decoding stages. The key strength is its training-free, dynamic approach that allows different tokens to be selected at different generation steps, achieving significant speedup (e.g., 2.34x for prefilling in multi-document QA) while maintaining accuracy. The paper's main weakness is the lack of comparison with parallel computation techniques, which are standard approaches for reducing TTFT. Additionally, the method introduces memory overhead from the Aux Cache during decoding, and the paper lacks detailed discussion on hyperparameter settings and implementation strategies. While the proposed method shows promise in accelerating inference, there are concerns about potential bias from attention-based pruning and the need for more comprehensive experimental validation.

**Additional Comments On Reviewer Discussion:**

During the rebuttal period, reviewers raised concerns about memory overhead, hyperparameter settings, and implementation details. The authors addressed these by clarifying that tokens are either in KV cache or Aux Cache (never both), leading to smaller total memory than baseline. They also provided additional implementation details including pseudocode and examples in the appendix. While some reviewers were satisfied with the responses (e.g., Reviewer 8NX8 maintained acceptance), others requested more comprehensive evaluation including end-to-end latency and throughput comparisons, which the authors provided in their final response.

---

### Decision · Program_Chairs · 2025-01-22

Reject